# Policy Message Passing: A New Algorithm for Probabilistic Graph Inference

## Abstract

A general graph-structured neural network architecture operates on graphs through two core components: (1) complex enough message functions; (2) a fixed information aggregation process. In this paper, we present the *Policy Message Passing* algorithm, which takes a probabilistic perspective and reformulates the whole information aggregation as stochastic sequential processes. The algorithm works on a much larger search space, utilizes reasoning history to perform inference, and is robust to noisy edges. We apply our algorithm to multiple complex graph reasoning and prediction tasks and show that our algorithm consistently outperforms state-of-the-art graph-structured models by a significant margin.

## 1 Introduction

Not every path is created equal. Powerful sequential inference algorithms have been a core research topic across many tasks that involve partial information, large search spaces, or dynamics over time. An early algorithm example is dynamic programming which memorizes the local information and uses a fixed inference trace to acquire a global optimum. In the modern deep learning era, sequential memory architectures such as recurrent neural networks are used for sequential inference (e.g. language generation (Sutskever et al. (2014))) to narrow down the search space. Neural-based iterative correction is developed in amortized inference for posterior approximation (Marino et al. (2018)). Also, a large bulk of works are developed in reinforcement learning for sequential inference of the environment with partial observation, non-future aware states, and world dynamics.

Graph-structured models are another class of models that heavily rely on local observations and sequential inference. It contains nodes and message functions with partial information, the final output of model relies on a sequence of communication operations which transforms the nodes information dynamically over iterations. A general formulation of learning graph-structured models is maximizing the likelihood:

$$\max_\theta p(\boldsymbol{y}|\boldsymbol{s_0}, \boldsymbol{x}; F_\theta, P) \tag{1}$$

The objective describes the process where it takes in the information $\boldsymbol{x}$ over a graph, starts from initial states $\boldsymbol{s_0}$ and maps to the target output $\boldsymbol{y}$. This process involves two key components, message functions $F$ and inference process $P$.

*Message function designs* focus on the choices of functions defined over the interaction between nodes or variables. Simple log-linear potential functions are commonly used in traditional probabilistic graphical models (Sutton et al. (2012)). Kernelized potential functions (Lafferty et al. (2004)), gaussian energy functions (Krähenbühl & Koltun (2011)), or semantic-driven potential functions (Lan et al. (2011)) are developed in varied cases to adapt to specific tasks. Deep graph neural networks generalizes the message functions to neural networks. There are a variety of function types developed such as graph convolution operations (Kipf & Welling (2016)), attention-based functions (Veličković et al. (2017)), gated functions (Li et al. (2015); Deng et al. (2016)).

*Inference procedures* over graphs are mainly developed under the context of probabilistic graphical models, with the classic inference techniques such as belief propagation ( Pearl (1988)), generalized belief propagation (Yedidia et al. (2001)), and tree-reweighted message passing ( Wainwright et al. (2003)). However, there has not been many research in developing powerful inference process in

modern deep graph-structured models (Graph Neural Networks). Most graph-structured models still follow a fixed hand-crafted rule for performing inference over nodes.

In this paper, we take a different tack and instead propose to represent the inference process modeled by a neural network $P_\phi$:

$$\max_{\phi,\theta} \mathbb{E}_{\boldsymbol{\tau} \sim P_\phi} \big[ p(\boldsymbol{y}|\boldsymbol{s_0}, \boldsymbol{x}, \boldsymbol{\tau}; F_\theta) \big] \tag{2}$$

The neural network $P_\phi$ represents the distribution over all possible inference trajectories in the graph model. By taking the distribution perspective of message passing, the whole inference is reformulated as stochastic sequential processes in the deep graph-structured models. This formulation introduces several properties: *Distribution nature* - the possible inference trajectories in graph, besides synchronous or asynchronous, are in a vast space with varied possibilities, leading to different results; *Consistent inference* - maintaining a powerful neural network memorizing over time could lead to more effective and consistent inference over time; *Uncertainty* - inference itself will affect prediction, with a distribution, such uncertainty over the reasoning between nodes can be maintained.

The primary contributions of this paper are three folds. Firstly, we introduce a new perspective on message passing that generalizes the whole process to stochastic inference. Secondly, we propose a variational inference based framework for learning the inference models. Thirdly, We empirically demonstrate that having a powerful inference machine can enhance the prediction in graph-structured models, especially on difficult reasoning tasks, or graphs with noisy edges.

## 2    BACKGROUND: GRAPH NEURAL NETWORKS

A large volume of Graph Neural Network (GNN) models have been developed recently (Defferrard et al. (2016); Li et al. (2015); Kipf & Welling (2016); Henaff et al. (2015); Duvenaud et al. (2015)) for varied applications and tasks. This line of methods typically share some common structures such as propagation networks and output functions. In our paper, without loss of generality, we follow a unified view of graph neural network proposed in a recent survey (Battaglia et al. (2018)) and introduce the graph neural network architecture here.

In this formulation, there are two key components: propagation networks and output networks. In the *propagation networks*, the fundamental unit is a "graph-to-graph" module. This module takes a graph with states $\boldsymbol{s}_t$ as input and performs information propagation (message passing) operations to modify the values to $\boldsymbol{s}_{t+1}$. The graph states at each step $t$ consists of sets of variables representing the attributes of nodes $\mathcal{V}_t = \{\boldsymbol{v}\}_t$ and edges $\mathcal{E}_t = \{\boldsymbol{e}\}_t$, along with global properties $\boldsymbol{u}_t$. The graph-to-graph module updates the attributes of these nodes, edges, and global properties via a set of message passing, implemented as neural networks.

$$\boldsymbol{e}_t = f_e\Big(\{\boldsymbol{v}\}_t, \mathcal{E}_t, E\Big); \boldsymbol{v}_t = f_v\Big(\{\boldsymbol{e}\}_t, \mathcal{V}_t, E\Big); \boldsymbol{u} = f_u\Big(\{\boldsymbol{v}\}_t, \{\boldsymbol{e}\}_t, E\Big) \tag{3}$$

The message passing in the standard graph neural networks is a fixed aggregation process defined by the graph structure $E$. The *output network* is a general part of model that maps the information in the graph states to the target output $y$: $p(\boldsymbol{y}|\boldsymbol{s}_T)$. In general, there is no restriction on the form of target $y$ or the output mapping function, like the target can be a set of labels for nodes, edges, the whole graph or even sentences.

## 3    GRAPH MODELS WITH POLICY MESSAGE PASSING

In this section we describe our stochastic graph inference algorithm under the formulation of graph neural networks. As the inference process under the algorithms is more of a reasoning procedure, we refer to it as reasoning in the later content. Our approach takes the initial state $\boldsymbol{s}_0$ of a graph representing observations and latent variables in a domain, uses a set of parameterized reasoning agents to pass messages over this graph, and infers the output prediction $\boldsymbol{y}$. Our model contains three main parts:

1. Graph $\boldsymbol{s}_t$, representing states of observed and latent variables.

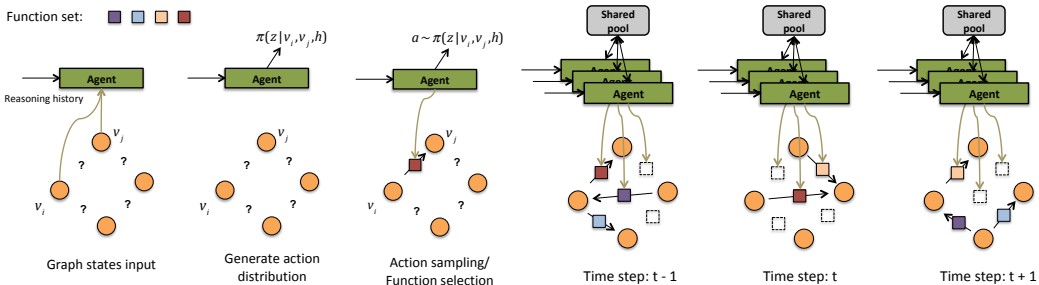

(a) Single timestep action proposal.          (b) Dynamic graph reasoning over multiple steps.

Figure 1: We design agents that learn to reason over graphs by sequentially performing actions for inference. (a) We learn a function set. Each function is a possible message along an edge. At each time step of inference, our reasoning agent examines the current graph states and produces a distribution over the function set to sample from. (b) A set of these agents, each one considering its own edge, is run to produce the full set of messages for a graph. Each agent maintains its own state and can communicate with the other agents via a shared memory.

2. A set of message functions $\{f_k(\cdot, \cdot; \theta_k)\}$, parameterized by $\theta_k$, that determine the possible manner in which messages can be passed over edges in the graph.

3. An automatic reasoning agent $A(\cdot; \phi)$, parameterized by $\phi$, that determines which message function would be appropriate to use along an edge in the graph.

For every timestep $t$ during reasoning, our model takes the current graph state $s_t$ and the previous reasoning history, then generates a distribution over actions for choosing which messages to pass (or pass no information). The new graph state $s_{t+1}$ is obtained through performing message passing actions sampled from the distribution. The final prediction is thus inferred sequentially through the operation of the reasoning agents over the graph. Fig. 1 illustrates the algorithm.

## 3.1 MODEL DEFINITION

Our inference procedure operates on graphs. We start with an initial graph state $s_0 = (v_0, e_0, u_0)$, where $v_0$, $e_0$, and $u_0$ contain information about individual nodes, edges, and global information about the graph respectively. Nodes in this graph can represent observations or latent variables from our domain; we assume the set of nodes is known and fixed. Edges in this graph represent possible relationships between nodes, over which we will potentially pass messages during inference. The edges are assumed known and fixed, though our method will permit flexibility via its inference.

Given a graph state $s$, we can output $y$ using a network $f_o(s; \theta_o)$. Note that in general this formulation could have a variety of possible output functions, corresponding to different tasks. In the current instantiation we focus on a single such output function that is used for a single task. Given the initial graph state $s_0$, we perform a set of reasoning steps that pass messages over this graph to obtain a refined graph representation. The goal of this message passing is to successively refine the graph state such that the graph state can be accurately decoded to match the correct output for the given input graph. The update functions we define will modify the node attributes $v$, though the formulation could be generalized to edge and global attributes as well.

**Message Formulation:** The message passing is decomposed into operations on edges in the graph. For the $i^{th}$ node in the graph, we update its state by passing messages from all nodes in its neighbour set $\mathcal{N}(i)$.

$$v^i = \sum_{j \in \mathcal{N}(i)} f_{z^{ij}}(v^i, v^j) \tag{4}$$

The form of message to pass is a crucial component of our method. We select the function $f_{z^{ij}}(\cdot, \cdot)$ to use in this message passing for each edge.

We define a common set of functions $\{f_k(\cdot, \cdot)\}_{0 \le k \le K}$ that can be used across any edge in the graph. To allow for no information to be passed along an edge, we define the function $f_0(\cdot, \cdot) = \mathbf{0}$. Each of the other functions $f_k(\cdot, \cdot)$ is a non-linear function (e.g. MLP) with its own learnable parameters $\theta_k$.

**Reasoning Agents:** Alongside these functions, we define a reasoning agent that determines which function to use on a particular edge in the graph at the current time step of inference. We run one copy of this reasoning agent for each edge of the graph. The reasoning agent takes as input the current graph state and for its edge $(i, j)$ outputs a distribution over the message to pass on that edge (computed by a network $A$). The reasoning agent maintains state $h_t^{ij}$ (propagated via a network $B$) over iterations of inference so that it can potentially remember which types of message passing actions were taken at different time steps $t$. We also include a global state $h_t^u = \sum_{ij} h_t^{ij}$ which can be used to share information across the agents operating on different edges.

The networks are updated and distributions computed using the following equations:

$$\boldsymbol{h}_{t+1}^{ij} = B(\boldsymbol{s}_t, i, j, \boldsymbol{h}_t^{ij}; \theta_B) \tag{5}$$

$$\pi(z_{t+1}^{ij}|\boldsymbol{s}_t) \propto A(\boldsymbol{s}_t, i, j, \boldsymbol{h}_t^{ij}, \boldsymbol{h}_t^u; \theta_A) + \mathbf{1} \tag{6}$$

where $\pi(z_{t+1}^{ij}|\boldsymbol{s}_t)$ is a distribution over $z_{t+1}^{ij}$, the message function to use for edge $(i, j)$ on step $t + 1$ given the current graph state. Note that a uniform prior is added to message distribution for regularization, though in general other priors could be used for this purpose. The parameters $\phi = (\theta_A, \theta_B)$ for the reasoning agent are shared; however, each edge maintains its own copy of these agents, via different states $h_t^{ij}$.

**Action sampling:** Given these distributions $\pi$, we can now use them to perform inference by sampling actions from these distributions. In our implementation, the Gumbel softmax trick is used for differentiable sampling.

By executing a set of actions $\{\boldsymbol{z}_t\}$ sampled from each of these distributions $\pi$, our model takes selected nodes and corresponding message functions to pass, and the graph state is updated to $\boldsymbol{s}_{t+1}$ after one reasoning/passing step. Over a sequence of time steps, this process corresponds to a reasoning trajectory $\boldsymbol{\tau} = (\boldsymbol{z}_1, \boldsymbol{z}_2, \ldots, \boldsymbol{z}_T)$ that produces a sequence of graph states $(\boldsymbol{s}_0, \boldsymbol{s}_1, \ldots, \boldsymbol{s}_T)$. At test time, we evaluate a learned model by sampling multiple such reasoning trajectory sequences and aggregating the results.

## 3.2 LEARNING OBJECTIVE

We assume that we are provided with training data that specify an input graph structure along with its corresponding value to be inferred. For ease of exposition, we derive our learning objective for a single training data pair of graph $\boldsymbol{s}$ with target value $\boldsymbol{y}$. For the learning objective, we adopt a probabilistic view on reasoning. As per the previous section, we can infer these values by running a particular reasoning process. However, rather than fixing a reasoning process, we could instead imagine inferring values $\boldsymbol{y}$ by marginalizing over all possible reasoning sequences $\boldsymbol{\tau}$.

$$p(\boldsymbol{y}|\boldsymbol{s}) = \sum_{\boldsymbol{\tau}} p(\boldsymbol{y}, \boldsymbol{\tau}|\boldsymbol{s}) = \sum_{\boldsymbol{\tau}} p(\boldsymbol{y}|\boldsymbol{\tau}, \boldsymbol{s})p(\boldsymbol{\tau}|\boldsymbol{s}) \tag{7}$$

The "prior" $p(\boldsymbol{\tau}|\boldsymbol{s})$ could be either a learnable prior or a hand-designed prior. Belief propagation or other neural message passing frameworks define a delta function where the prior is deterministic and fixed. Here we instead adopt a learnable prior $\pi$ for $p(\boldsymbol{\tau}|\boldsymbol{s})$, where the reasoning process depends on the graph states. $\boldsymbol{\tau}$ represents the latent reasoning sequence. Given a graph initial state and ground truth label $\boldsymbol{y}$, we are interested in the posterior distribution over trajectories that describes an efficient and effective reasoning process. We adopt standard variational inference to approximate the posterior distribution and try to find such action spaces.

As per the previous section, we have a method for computing $\boldsymbol{y}$ given a reasoning sequence $\boldsymbol{\tau}$. However, as in variational inference applications, evaluation of the summation in Eq. 7 is challenging. Standard sampling approaches, i.e. choosing an inference procedure from an arbitrary proposal distribution, would be likely to lead to inference procedures ineffective for the given graph/inference task. Instead, we utilize our reasoning agent as a proposal distribution to propose inference procedures. If the proposal distribution is close to the posterior distribution over reasoning actions, then we can

conduct both effective and efficient inference to obtain the final desired output. Hence, we hope to minimize the following KL divergence:

$$KL(q(\boldsymbol{\tau})||\pi^*(\boldsymbol{\tau}|\boldsymbol{y}, \boldsymbol{s})) \tag{8}$$

where $q$ is the proposal distribution over reasoning actions. $\pi^*$ is the true posterior distribution over reasoning sequences given the target inference values $\boldsymbol{y}$ and graph state $\boldsymbol{s}$, which is unknown. We adopt the standard variational inference method to derive the following equations:

$$\log p(\boldsymbol{y}|\boldsymbol{s}) - KL(q(\boldsymbol{\tau}|\boldsymbol{y}, \boldsymbol{s})||\pi^*(\boldsymbol{\tau}|\boldsymbol{y}, \boldsymbol{s})) = \mathbb{E}_{\boldsymbol{\tau} \sim q}\big[\log p(\boldsymbol{y}|\boldsymbol{\tau}, \boldsymbol{s})\big] - KL(q(\boldsymbol{\tau}|\boldsymbol{s}, \boldsymbol{y})||\pi(\boldsymbol{\tau}|\boldsymbol{s})) \tag{9}$$

We can optimize the right hand side, which is also known as the evidence lowerbound (ELBO). The log likelihood and KL divergence in Eq. 8 are then implicitly optimized. We can also decide to define the learning objective as a product of probabilities over the $T$ time steps in the reasoning process. In this case, the objective would be:

$$\left[\sum_{t=1}^{T} \mathbb{E}_{\boldsymbol{z}_t \sim q}\big[\log p(\boldsymbol{y}|\boldsymbol{z}_t, \boldsymbol{s}_{t-1})\big]\right] - KL(q(\boldsymbol{\tau}|\boldsymbol{s}_0, \boldsymbol{y})||\pi(\boldsymbol{\tau}|\boldsymbol{s}_0)) \tag{10}$$

To summarize, we learn the parameters of the networks $f_o$ and $\{f_k\}$ which specify $p(\boldsymbol{y}|\boldsymbol{\tau}, \boldsymbol{s})$; and $A$ and $B$ which specify $\pi(\boldsymbol{\tau}|\boldsymbol{s})$. This is accomplished by maximizing the ELBO in Eq. 10 over all graph-target value pairs in the training data.

**Interpretation of Learning Objective:** The learning objective consists in two terms. *Likelihood term:* This term provides main objective for the agent and model to maximize. The policy should be learned to perform accurate reasoning to fit the data and maximize the likelihood of the ground truth values for $\boldsymbol{y}$. *KL term:* The policy we learn in proposal distribution can be used to guide the learnable prior agent toward good inference procedures. The KL divergence term in the objective function can be used to achieve this.

### 3.3 MIXED SAMPLING FROM SEQUENTIAL DISTRIBUTIONS

As the whole inference over graphs is a sequential procedure, there is a common issue that will appear in the above formulation: the drifting problem when facing unfamiliar graph states in some inference step $t$. The proposal distribution in the variational inference formulation has access to ground truth data, which will lead to "good" graph states in most cases, while the prior agent will probably face some graph states that will lead to failure predictions and even drift away further. Similar issues have been found in sequential data generation (Lamb et al. (2016)) or imitation learning (Ross et al. (2011)). In our model, we adopt a easy fix by mix the sampling from both prior and proposal in the trajectory level, to have graph neural networks visit different states, and by firstly rollout using prior agent, then use the proposal agent to complete the rest trajectory. The objective is re-weighted using importance sampling weights when mixing the trajectories.

## 4 EXPERIMENTS

We test our proposed model on two types of modalities of data: complex visual reasoning tasks and social network classification. Visual data has more information and more complex possible interactions or variations between nodes in the graph, while social network has larger scale of graphs.

### 4.1 VISUAL REASONING

We introduce two visual reasoning tasks: (1) visual cue-based position reasoning and (2) large-scale image puzzle solving. To solve these visual based reasoning tasks, we require a model to have a strong ability in handling imperfect observations, performing inference, handling noisy edges, and flexibly making modifications to graph nodes.

**Baseline methods:** We compare our proposed model with several popular graph neural network based variants. These models all more or less have the ability to learn and control the information flow, and have been demonstrated to have strong performance on many tasks or applications.

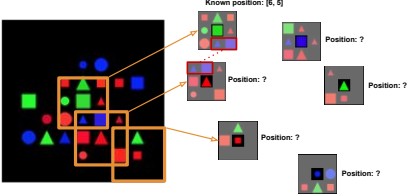

| number of objects | 25 | 36 |
|---|---|---|
| random | 4 | 2.78 |
| GNN | 32.92 | 16.21 |
| GNN-attention | 40.26 | 21.58 |
| RRN | 35.10 | 19.62 |
| NRI | 38.74 | 18.84 |
| PMP (ours) | **81.26** | **46.8** |

(a) Illustration of the *"Where am I"* dataset and tasks. Each object is extracted from the original image along with its context. Between some patches there will be similar patterns appear.

(b) Performance results on *"Where am I"* dataset, measured by per-node position classification accuracy.

Figure 2: Our "Where am I" dataset(left) and results compared with baseline methods measured by per object accuracy(right).

- *Graph Neural Network* (Battaglia et al. (2018)). We follow the graph neural network model described in the survey Battaglia et al. (2018). The graph neural network contains three update functions: node, edge, global states. The aggregation process we used is mean pooling of messages.

- *Attention-based Graph Neural Network* (Veličković et al. (2017); Wang et al. (2018); Vaswani et al. (2017)). An attention mechanism is used for modulating the information flow from different messages and can be viewed as a non-probabilistic single-step version of our model. We implement the attention by using sigmoid gates over different message functions.

- *Recurrent Relational Network* (Battaglia et al. (2018)). We follow the open source code and implement the recurrent relational network (RRN) model. The RRN model updates the node states through a Gated Recurrent Unit and adds dense supervision across all timesteps.

- *Neural Relational Inference model* (Kipf et al. (2018)). The Neural Relational Inference model is the first proposed probabilistic model that captures the distribution over graph structures. We follow the same implementation and use four function types on each edge with entropy as an extra penalty in the objective function.

### 4.1.1   *"Where am I"*: VISUAL POSITION REASONING

**Task design:** We firstly design a synthetic dataset named *"Where am I"*. The dataset is based on the Shapes Andreas et al. (2016) dataset, initially proposed in the Visual Question Answering (VQA) context. The dataset contains 30K images with shape objects randomly dropped on $k \times k$ grids. We assume that each shape object has a $3 \times 3$ context field of view and ensure that all those shape objects are connected within the visible context. The dataset is set up to have two grid settings, $9 \times 9$ and $12 \times 12$, which contain 25 objects and 36 objects respectively. For each image, the position for one object is known and the model needs to perform inference over objects by comparing the context to infer the correct position of other objects.

**Model setup and problem difficulties:** We build a graph to solve this problem. Each node in the graph represents one object, one of them contains a known position tuple (x, y), and the positions for the rest obejcts are unknown. The graph is assumed to be fully connected. We use graph networks with 50 hidden dimension on both node and edge, ELU as activation functions. Each model takes 7 and 8 steps of inference for 25 object and 36 object cases respectively. The difficulty of this problem is that for each object, there are 16 ways/configurations to overlap with another object. Each position configuration can contain $N$ candidates. If $N$ is larger than 1, there will be confusion, and models need to solve other objects first to provide more context information. The task in general needs model to perform iterative modification of nodes content and avoid false messages.

**Experiment results:** Table 2b shows the performance of each model measured by the classification accuracy for each object. Our proposed Policy Message Passing (PMP) model and its variants achieve a large performance boost over the baselines, demonstrating that there exists a strong need for this kind of dynamical reasoning in complex tasks. The models with fixed structure and rigid aggregation

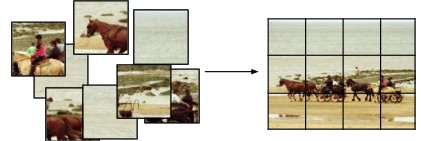

(a) Illustration of the image puzzle solving task. The example image is from visual genome.

| Models | Acc(%) | | | | Kendall-tau | | | |
|---|---|---|---|---|---|---|---|---|
| | 2x2 | 3x3 | 4x4 | 5x5 | 2x2 | 3x3 | 4x4 | 5x5 |
| Mena et al. (2018) | 1.0 | .97 | .9 | .79 | 1.0 | .96 | .88 | .78 |
| PMP | **1.0** | **1.0** | **.99** | **.99** | **1.0** | **1.0** | **.99** | **.99** |

(b) We compare our model's performance with state-of-the-art deep learning based puzzle solving model on Celeba dataset.

Figure 3: Image puzzle solving problem(left) and results compared with state-of-the-art method on Celeba dataset(right).

processes tend to not learn or generalize well. Attention based models slightly alleviate that problem, but are not able to perform consistent and strategic reasoning over objects.

### 4.1.2 IMAGE PUZZLE SOLVING ON BENCHMARKS

**Task design and datasets:** Next we move on to large-scale benchmark datasets with real images containing complex visual information. We test our model on the Jigsaw Puzzle solving task for real images. Given an image with size $H \times W$, we divide the image into $d \times d$ non-overlapping patches. The $d^2$ patches are un-ordered and need to be reassembled into the original image. We use two datasets to mainly test our model: (1) Visual Genome - The Visual Genome dataset contains 64346 images for training and 43903 images for testing. It is often used in image based scene graph generation tasks, indicating that there exists very rich interactions and relationships between visual elements in the image. We test the models on three different puzzle sizes: $3 \times 3, 4 \times 4$, and $6 \times 6$ (2) COCO dataset - We use the COCO dataset (2014 version) that contains 82783 images for training and 40504 images for testing. We set up three tasks, $3 \times 3, 4 \times 4$ and $6 \times 6$ for evaluation.

**Evaluation metric and model implementation:** To evaluate the models' performance, we use per-patch classification score over the $d^2$ classes and measure the Kendall-tau correlation score between predicted patch order and the ground-truth order. For all graph models, we use 50 hidden dimensions on both nodes and edges. Models will perform $d + 2$ steps of inference. For all baseline methods, we use 200 dimensions for nodes and messages, while still remain 50 dimensions for our main model.

| Models | Accuracy(%) | | | Kendall-tau | | |
|---|---|---|---|---|---|---|
| | 3x3 | 4x4 | 6x6 | 3x3 | 4x4 | 6x6 |
| random | 11.11 | 6.25 | 2.78 | - | - | - |
| GNN | 93.86 | 77.37 | 50.81 | 94.54 | 82.54 | 74.55 |
| GNN-attention | 91.29 | 81.36 | 68.39 | 92.70 | 86.56 | 79.11 |
| RRN | 89.91 | 78.14 | 62.52 | 85.33 | 84.46 | 76.63 |
| NRI | 87.63 | 81.93 | 69.01 | 89.12 | 86.67 | 79.78 |
| PMP | **98.31** | **97.96** | **90.53** | **99.27** | **98.67** | **96.01** |

Table 1: Performance results on COCO dataset, measured by per-patch classification accuracy and correlation score Kendall-tau. We use Kendall-tau$\times 100$ here for better comparison. Results are shown across different puzzle sizes.

**Experiment results:** Tables 1 and 4a summarize the results of all models across varied task settings. Our proposed models consistently outperform baseline models by a $10\%$ - $20\%$ margin, even when baseline models have much larger capacity on the graphs. This is a further indication that having a powerful inference process is crucial to deep graph-structured models.

**Analysis on number of inference steps and size of function set:** We further examine the impact of inference steps and number of functions in function set on our models, shown in 4b. We found that empirically a function set of size 4 is enough for message diversity. For inference step analysis, we test our PMP model on $6 \times 6$ puzzles and found with 6 steps the model can already achieve a good performance. Similar behaviours are found for baseline models too. We only plot our main model in the figure.

| Models | Accuracy(%) | | | Kendall-tau | | |
|---|---|---|---|---|---|---|
| | 3x3 | 4x4 | 6x6 | 3x3 | 4x4 | 6x6 |
| random | 11.11 | 6.25 | 2.78 | - | - | - |
| GNN | 82.83 | 72.46 | 61.22 | 85.47 | 79.94 | 79.58 |
| GNN-attention | 86.87 | 77.35 | 71.27 | 88.75 | 85.51 | 83.35 |
| RRN | 85.60 | 77.27 | 69.23 | 88.07 | 83.31 | 78.50 |
| NRI | 87.63 | 73.51 | 70.62 | 89.37 | 80.06 | 79.91 |
| PMP | **97.92** | **96.74** | **94.61** | **98.58** | **98.07** | **97.05** |

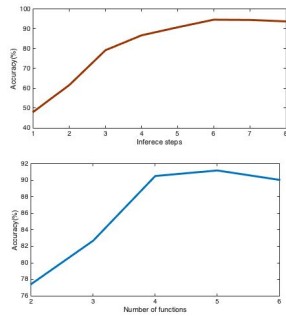

(a) Performance results on Visual Genome dataset, measured by per-patch classification accuracy and correlation score Kendall-tau. Results are shown across different puzzle sizes.

(b) Analysis of our model on number of inference steps and number of functions.

Figure 4: Performance on Visual Genome(left) and model analysis plots(right).

**Comparison with state-of-the-art puzzle solving method**: We compared our method with state-of-the-art deep learning based image puzzle solving on the CelebA dataset. The results are shown in Table 3b. Our model shows near perfect puzzle solving results on $2 \times 2, 3 \times 3, 4 \times 4$ and $5 \times 5$ settings.

## 4.2 SOCIAL NETWORK CLASSIFICATION WITH NOISY EDGES

We further inspect on models' ability on handling noisy edges on large scale graphs. Current graph neural network benchmark datasets on social networks, such as Cora (Sen et al. (2008)), heavily rely on well caliberated edge matrix as a strong prior information for inference. The burden on aggregation process are less, compare to more general cases where adjacency matrix information are often noisy or missing. In this experiment, we compare our model's performance with two state-of-the-art graph-structured models: (1) Graph Convolutional Networks (GCN) and (2) Graph Attention Networks (GAT) on a standard benchmark Cora. Instead of using only the well designed adjacency matrix information, we also generate perturbation to the matrix by adding $\{\%50, \%100, \%200\} * |E|$ more noisy edges, $|E|$ is the number of edges in the original graph. For implementation, we use the official release of the codes on two baseline models and default settings. In our proposed model, we follow the same hyper-parameters in graph networks and uses recurrent network agent with 32 hidden dimensions. Results summarized in table 2 show that our model is more robust to noisy edges in performing inference over large scale graphs.

| Models | Noisy edges | | |
|---|---|---|---|
| | 50% | 100% | 200% |
| GCN | $26.55 \pm 1.3$ | $23.53 \pm 1.44$ | $22.62 \pm 1.84$ |
| GAT | $58.83 \pm 1.35$ | $45.82 \pm 2.55$ | $34.53 \pm 0.45$ |
| PMP | $\mathbf{65.37 \pm 1.36}$ | $\mathbf{61.42 \pm 1.31}$ | $\mathbf{50.3 \pm 1.13}$ |

Table 2: The performance of node classification for Cora dataset under varied noisy edge settings. The experiment follows the standard transductive setup in Yang et al. (2016).

## 5 CONCLUSION

Reasoning over graphs can be formulated as a learnable process in a Bayesian framework. We described a reasoning process that takes as input a graph and generates a distribution over messages that could be passed to propagate data over the graph. Both the functional form of the messages and the parameters that control the distribution over these messages are learned. Learning these parameters can be achieved via a variational approximation to the marginal distribution over provided graph label. Future possibilities include adapting our model to use a more general set of functions where some of them is not differentiable.

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
