# OpenReview forum: "Policy Message Passing: A New Algorithm for Probabilistic Graph Inference"
_ICLR.cc/2020/Conference — Reject_

### Official Review · AnonReviewer1 · 2019-10-08
**Official Blind Review #1**

**Rating:** 1

**Review:**

This paper introduces policy message passing: a graph neural network with multiple message types and an inference mechanism that assigns messages of a particular type to edges in a recurrent fashion. Experiments on visual reasoning tasks and on a noisy citation network classification task indicate competitive performance.

There are many interesting ideas in this paper and I believe that the overall approach/idea has merit and is interesting for the community. In its current state, however, the paper has issues in terms of clarity of writing, technical and notational mistakes, unclear experimental settings and unclear implementation of baseline models, and needs a major revision, hence I recommend reject. This feedback is mostly meant as an encouragement to “polish” the ideas outlined in the paper and their presentation, as I believe that these can be quite impactful, if presented well and once all technical issues have been addressed.

Discussion of the main idea outlined in this paper:
———————————————————————
The proposed dynamic message passing procedure (termed “reasoning agents” in the paper) can be understood as an extension of Neural Relational Inference (NRI; Kipf et al., ICML 2018), where a Gumbel softmax-based attention mechanism in parallel softly assigns a particular message function per edge in the graph. The main difference to NRI is that this is done sequentially over multiple message passing steps in the graph, whereas NRI (in the training phase) only performs one single such step (for all edges in the graph).

Getting this to work well in practice is a challenging problem, as sequential sampling steps can accumulate biases in the approximate inference scheme. The authors propose to address this issue in a number of ways: the attention mechanism per edge is coupled over time steps via a local hidden state and coupled to all other edges via a global hidden state (this is novel), a vector of ones is added to the action probabilities and subsequently re-normalized to act as a regularizer (this appears to be a hack), they use a learned prior distribution instead of a fixed prior as in NRI, and lastly, the authors introduce a mixed sampling strategy which mixes samples from the (learned) prior distribution for edge types and the proposal distribution while re-weighing the objective with importance sampling weights — for this last contribution (mixing + reweighing) the description in the paper misses some details (how precisely is the mixing done over the course of training?).

Clarity, technical correctness, experimental evaluation
———————————————————————————
* The paper contains a lot of spelling mistakes and grammatical errors, making it difficult to read.
* Notational issues: Eq 3 is unclear (no node/edge indices provided, function arguments unspecific — all three functions have the same arguments); Eq. 8/9 switches between posteriors p, true posteriors \pi^* (what’s the difference from p?) and priors \pi without explanation— a derivation would be helpful. Eq. 10 only places a prior on the first time step — where does this come from? A derivation would be helpful.
* “Belief propagation or other neural message passing frameworks define a delta function where the prior is deterministic and fixed” -> this is not the case for NRI
* What temperature is used for the Gumbel softmax distribution? Does the forward pass use “hard” (straight-through trick) or soft activations?
* Eq. 5: Why condition B on the indices i and j? This will likely not generalize to new edges
* Eq. 6: What is the motivation for adding a vector of ones? Wouldn't it be possible to achieve the desired regularization effect by increasing the softmax temperature?
* How are the edge and global hidden states initialized?
* The explanation for the mixed sampling procedure is unclear, please elaborate (potentially use an appendix)
* What are the train/val/test splits, how are hyper parameters optimized in all experiments?
* Why implement attention-based GNN baselines with sigmoid gates even though you cite Transformers and Graph Attention Networks (GAT) in this paragraph? I suspect that multi-head dot product attention (default for Transformers/GAT) instead of sigmoid gating would perform much better.
* Please explain the architecture / experimental setting used in the GNN and RRN baselines in an appendix. Please also clarify how you use NRI (recurrent version, Markovian version etc.)
* “Where am I” task: the model setup paragraph is unclear
* Figure 3a: the example provided is not d x d (but 3x4 in terms of # puzzle pieces)
* Ablation studies: please consider including experiments that analyze how the proposed model performs without global state, without the added prior (vector of ones), without the mixed sampling procedure, without recurrence and with different numbers of functions per edge.


Other comments:
* Please consider familiarizing yourself with the \citep command in LaTeX for parenthetical citations to avoid sequences of closing parentheses.


**Experience Assessment:**

I have published in this field for several years.

**Review Assessment: Checking Correctness Of Derivations And Theory:**

I assessed the sensibility of the derivations and theory.

**Review Assessment: Checking Correctness Of Experiments:**

I assessed the sensibility of the experiments.

**Review Assessment: Thoroughness In Paper Reading:**

I read the paper thoroughly.

---

### Official Review · AnonReviewer2 · 2019-10-26
**Official Blind Review #2**

**Rating:** 3

**Review:**

The paper presents a policy message passing algorithm for graph neural network where both node & edge energy functions and the message passing procedure are treated in a unified manner as outputs of neural nets. In particular, the message passing procedures for individual edges are individualized (i.e. not homogeneous) while a common memory pool is also shared among these procedures. It also aims at doing message passing in the distribution level instead of singling out a particular message passing rule. The idea is new (to my understanding) and interesting. Empirical results demonstrate the usefulness of the proposed methods. On the other hand, the main concerns are:
*Only a rough idea is introduced and discussed, then followed by some nice practical results. In the middle, there is lacking of concrete execution of the proposed idea. The devil could lie in the details.
*many details are missing. Throughout the paper, it is not clear at all how the message functions and the inference procedure functions (or called reasoning agents) are realized in practice. Also I do not see the practical implementation would be made publicly accessible.


**Experience Assessment:**

I have read many papers in this area.

**Review Assessment: Checking Correctness Of Derivations And Theory:**

I assessed the sensibility of the derivations and theory.

**Review Assessment: Checking Correctness Of Experiments:**

I assessed the sensibility of the experiments.

**Review Assessment: Thoroughness In Paper Reading:**

I read the paper at least twice and used my best judgement in assessing the paper.

---

### Official Review · AnonReviewer3 · 2019-10-26
**Official Blind Review #3**

**Rating:** 3

**Review:**

This paper proposes Policy Message Passing, a Bayesian GNN which models edge message passing as a mixture distribution with corresponding coefficient generated by a learnable prior defined on graph state. I think the motivation is reasonable and the experiment on Jigsaw Puzzle is impressive. But the writing and other experiments can be further improved, as detailed below.

Sec 3.1: What does ```````+1 mean in Eq. (6)?

Sec 3.2: The choice for the variational distribution q is not mentioned. Is q a parametric model (e.g., NN) or non-parametric? It is hard to connect prior, posterior and variational posterior in Sec 3.2 to the reasoning agent/message formulation in Sec 3.1. I suggest reformulating Sec 3.1 & 3.2 to give a thorough presentation of the training objective, model structure and optimization details of PMP.

Modeling graph using graph neural network with Bayesian framework has been investigated in several papers, e.g., [1-2]. They also take a probabilistic perspective. I think this paper should discuss the connections to previous Bayesian GNNs and compare their performance with PMP at least in Sec 4.2. Besides, GNN with edge information (e.g., [3]) has also been investigated in several papers. According to my understanding, PMP can also be regarded as a GNN with latent edge information (modeled with a learnable prior and inferred with a variational posterior). So I think adding discussion/experiment to them can better support your claim.

As for the experiment part, my major concern is whether the comparison with other models (e.g., GCN and GAT) is fair. It seems that the number of parameters of PMP is much larger than GCN and GAT. Could you provide details for the model size and the training/inference time cost?

Sec 4.2. What's the performance of PMP when no noisy edges is added?

[1] Bayesian graph convolutional neural networks for semi-supervised classification, AAAI 2019
[2] Bayesian Semi-supervised Learning with Graph Gaussian Processes, NIPS 2018
[3] Exploiting Edge Features for Graph Neural Networks, CVPR 2019


**Experience Assessment:**

I have read many papers in this area.

**Review Assessment: Checking Correctness Of Derivations And Theory:**

I carefully checked the derivations and theory.

**Review Assessment: Checking Correctness Of Experiments:**

I carefully checked the experiments.

**Review Assessment: Thoroughness In Paper Reading:**

I read the paper thoroughly.

---

### Decision · Program_Chairs · 2019-12-19

**Decision:**

Reject

**Comment:**

This paper was reviewed by 3 experts, who recommend Weak Reject, Weak Reject, and Reject. The reviewers were overall supportive of the work presented in the paper and felt it would have merit for eventual publication. However, the reviewers identified a number of serious concerns about writing quality, missing technical details, experiments, and missing connections to related work. In light of these reviews, and the fact that the authors have not submitted a response to reviews, we are not able to accept the paper. However given the supportive nature of the reviews, we hope the authors will work to polish the paper and submit to another venue.